# Diagnostic Potential of *miR-143-5p, miR-143-3p, miR-551b-5p,* and *miR-574-3p* in Chemoresistance of Locally Advanced Gastric Cancer: A Preliminary Study

**DOI:** 10.3390/ijms25158057

**Published:** 2024-07-24

**Authors:** Marlena Janiczek-Polewska, Tomasz Kolenda, Paulina Poter, Joanna Kozłowska-Masłoń, Inga Jagiełło, Katarzyna Regulska, Julian Malicki, Andrzej Marszałek

**Affiliations:** 1Department of Clinical Oncology, Greater Poland Cancer Centre, 61-866 Poznan, Poland; 2Department of Electroradiology, Poznan University of Medical Sciences, 61-701 Poznan, Poland; 3Laboratory of Cancer Genetics, Greater Poland Cancer Centre, 61-866 Poznan, Poland; 4Research and Implementation Unit, Greater Poland Cancer Centre, 61-866 Poznan, Poland; katarzyna.regulska@wco.pl; 5Department of Clinical Pathology, Poznan University of Medical Sciences and Greater Poland Cancer Centre, 61-866 Poznan, Poland; 6Institute of Human Biology and Evolution, Faculty of Biology, Adam Mickiewicz University, 61-614 Poznan, Poland; 7Pharmacy, Greater Poland Cancer Centre, 61-866 Poznan, Poland; 8Department of Clinical Pharmacy and Biopharmacy, Poznan University of Medical Sciences, Collegium Pharmaceuticum, 60-806 Poznan, Poland

**Keywords:** microRNAs, locally advanced gastric cancer, pathological staging, clinical staging, neoadjuvant chemotherapy

## Abstract

Gastric cancer (GC) is one of the most frequently diagnosed cancers in the world. Although the incidence is decreasing in developed countries, the treatment results are still unsatisfactory. The standard treatment for locally advanced gastric cancer (LAGC) is gastrectomy with perioperative chemotherapy. The association of selected microRNAs (miRNAs) with chemoresistance was assessed using archival material of patients with LAGC. Histological material was obtained from each patient via a biopsy performed during gastroscopy and then after surgery, which was preceded by four cycles of neoadjuvant chemotherapy (NAC) according to the FLOT or FLO regimen. The expression of selected miRNAs in the tissue material was assessed, including *miRNA-21-3p*, *miRNA-21-5p*, *miRNA-106a-5p*, *miRNA-122-3p*, *miRNA-122-5p*, *miRNA-143-3p*, *miRNA-143-5p*, *miRNA-203a-3p*, *miRNA-203-5p*, *miRNA-551b-3p*, *miRNA-551b-5p*, and *miRNA-574-3p*. miRNA expression was assessed using quantitative reverse transcription polymerase chain reaction (qRT-PCR). The response to NAC was assessed using computed tomography of the abdomen and chest and histopathology after gastrectomy. The statistical analyses were performed using GraphPad Prism 9. The significance limit was set at *p* < 0.05. We showed that the expression of *miR-143-3p*, *miR-143-5p*, and *miR-574-3p* before surgery, and *miR-143-5p* and *miR-574-3p* after surgery, decreased in patients with GC. The expression of *miR-143-3p*, *miR-143-5p*, *miR-203a-3p*, and *miR-551b-5p* decreased in several patients who responded to NAC. The miRNA most commonly expressed in these cases was *miRNA-551b-5p*. Moreover, it showed expression in a patient whose response to chemotherapy was inconsistent between the histopathological results and computed tomography. The expression of *miR-143-3p*, *miR-143-5p*, *miR-203a-3p*, and *miR-551b-5p* in formalin-fixed paraffin-embedded tissue (FFPET) samples can help differentiate between the responders and non-responders to NAC in LAGC. *miR-143-3p*, *miR-143-5p*, and *miR-574-3p* expression may be used as a potential diagnostic tool in GC patients. The presence of *miR-551b-5p* may support the correct assessment of a response to NAC in GC via CT.

## 1. Introduction

Gastric cancer (GC) is the fifth most common cancer in terms of incidence and the fourth most common cause of death [1]. In 2020, approximately 1.1 million new cases of stomach cancer were diagnosed, and approximately 800,000 deaths from this cause were recorded [1]. The incidence and mortality of GC are decreasing in developed countries, although it is still a significant problem in East Asia and Europe [1,2]. Additionally, an increase in incidence in people of <50 years of age has been recently demonstrated [3]. Despite progress in the treatment of patients with GC, the results are still unsatisfactory [4]. According to the TNM and Union for International Cancer Control (UICC) classification, endoscopic or surgical R0 resection is recommended in the early stages of GC. This is the only treatment option that allows for a cure. The 5-year overall survival (OS) rate ranges from 93.6% to 94.2% [5]. Patients with early-stage GC may be asymptomatic. Only patients with advanced GC may experience weight loss, abdominal pain, vomiting, dysphagia, and upper gastrointestinal bleeding. Therefore, patients with advanced GC are more often diagnosed [6,7]. The standard treatment for locally advanced GC (LAGC) is perioperative chemotherapy [8,9,10,11]. The first breakthrough study that confirmed the effectiveness of perioperative chemotherapy was the MAGIC (Medical Research Council Adjuvant Gastric Infusional Chemotherapy) study published in 2006. It was shown that perioperative chemotherapy with epirubicin, cisplatin, and fluorouracil or capecitabine (i.e., ECF/ECX regimens) improved overall survival (OS) by 13% compared to surgery alone [8]. This was followed by subsequent studies demonstrating the superiority of neoadjuvant docetaxel, oxaliplatin, fluorouracil, and leucovorin (FLOT) over ECF/ECX regarding pathological response and OS. These studies are the basis for the current perioperative chemotherapy standards in locally advanced GC [9,10]. Although perioperative chemotherapy has become the standard for patients with LAGC, only 50–65% of patients who undergo neoadjuvant chemotherapy followed by surgical removal receive postoperative chemotherapy [8,9,12]. Chemotherapy, according to the FLOT regimen, carries many complications, which has led to a reflection on the use of a greater selection of patients with LAGC for perioperative chemotherapy. Van Putten et al. demonstrated improved OS in patients who underwent perioperative treatment compared with those who underwent preoperative treatment [13]. On the other hand, in a more recent analysis, the median OS was similar in patients who received adjuvant chemotherapy (AC) and also in those who did not [14]. Consideration should also be given to patients who do not benefit at all from perioperative chemotherapy, that is, patients who do not respond to neoadjuvant chemotherapy. One of the main causes for this is the development of drug resistance, which results in the failure of chemotherapy. Drug resistance can be divided into innate and acquired. Resistance mechanisms include the inhibition of cell apoptosis, changes in the cell cycle, changes in drug efflux, enhanced DNA damage repair, and the dysregulation of epithelial–mesenchymal transformation (EMT). However, the detailed mechanisms involved in drug resistance are still unknown [15]. The antitumor effects of cytotoxic drugs are extensive and without high selection, but the correlation between genetic traits and chemosensitivity may also be underestimated. Polymorphisms, gene mutations, and unique genetic backgrounds may lead to different rates of response to the same chemotherapy regimen [16]. Multidrug resistance (MDR) is one of the most significant reasons for chemotherapeutic failure in gastric cancer. Although accumulating investigations and research have been performed to elucidate the mechanisms of multidrug resistance, the details are far from being completely understood. The importance of microRNAs in cancer chemotherapeutic resistance has recently been demonstrated, providing a new strategy for overcoming multidrug resistance. The different mechanisms are related to the phenomenon of MDR itself and the roles of miRNAs in these multi-mechanisms by which MDR is acquired [17]. MicroRNAs (miRNAs) are a class of small noncoding nucleic acids. miRNAs act as master regulators in the control of gene expression. Currently, we know of over 2600 human-specific miRNAs [18,19,20,21]. Dysregulation is associated with apoptosis, cancer cell proliferation, invasion, and metastasis [22,23,24]. miRNA expression patterns can be successfully assessed in various samples, including biopsy, resection specimens, or blood samples, which supports the hypothesis that miRNAs could be used as clinically relevant diagnostic or therapeutic molecular biomarkers. Formalin-fixed paraffin-embedded tissue (FFPET) samples are an invaluable source for the study of human disease. Many tissue blocks are archived worldwide with corresponding well-documented clinical histories and histopathological reports, providing an easily accessible and non-invasive source of data [25,26]. Within the scope of the current research, we anticipate being able to preliminarily identify the occurrence of miRNAs in LAGC. Understanding this phenomenon could help us comprehend the mechanism of chemoresistance in LAGC. However, it is worth emphasizing that identifying the relevant miRNAs in LAGC, both before and after surgery, will not exclude the successful implementation of perioperative chemotherapy as an efficient method; rather, it will indicate any possible limitations to introducing chemotherapy in some cases and will enable the effective recruitment of patients to undergo systemic therapy. In addition, it will allow for the administration of chemotherapy in chemoresistant patients to be limited, making it possible to eliminate unnecessary toxicity during treatment. Thus, new knowledge on tools for the assessment of chemoresistance in LAGC is crucial, and the obtained results may be of great importance for developing clinical oncology. Our study aimed to evaluate the role of our selected miRNAs in chemoresistance in LAGC. The selection of miRNAs was based on the available literature [27,28]. We considered miRNAs that were associated with chemoresistance in GC or associated with GC progression, and we assessed them in FFPET. Our research is preliminary in nature and provides a foundation for further study. It is also innovative, as this type of research has not been found in the existing literature.

## 2. Results

### 2.1. Expression Levels of miRNA in GC Patients Based on the Cancer Genome Atlas

Using data from The Cancer Genome Atlas (TCGA), the expression of selected miRNAs was determined based on the expression levels of the *miRNA-21-3p*, *miRNA-21-5p*, *miRNA-106a-5p*, *miRNA-122-3p*, *miRNA-122-5p*, *miRNA-143-3p*, *miRNA-143-5p*, *miRNA-551b-3p*, *miRNA-551b-5p*, and *miRNA-574-3p* in 372 patients with GC and in 32 controls with normal stomach tissue. The data showed that five of our selected miRNAs are expressed in GC. We considered an FDR of <0.25 and a *p*-value of <0.05. According to the ENCORI database, we observed the up-regulation of *miRNA-21-3p* (*p =* 1.83 × 10^−68^), *miRNA-21-5p* (*p =* 6.0 × 10^−24^), and *miRNA-106a-5p* (*p =* 0.00099), as well as a down-regulation of *miRNA-143-5p*, in cancer patients than in the normal controls. No differences were noted for *miRNA-122-3p*, *miRNA-122-5p*, *miRNA-143-3p*, *miRNA-551b-3p*, *miRNA-551b-5p*, and *miRNA-574-3p.*

Moreover, no information was found in the database about *miRNA-203a-3p* and *miRNA-203a-5p*. Only the graph included in the graphic was found. All data are presented in Table 1.

Next, using the UALCAN database, we checked the differences in the expression levels of *miRNA-21*, *miRNA-106a*, *miRNA-122*, *miRNA-143*, *miRNA-203a*, *miRNA-551b*, and *miRNA-574* depending on the clinicopathological parameters in the GC patients included in the TCGA project. First of all, we checked the expression levels of those miRNAs in primary tumors (n = 387) in comparison to normal samples (n = 40). We observed the up-regulation of *miRNA-21* (284,544.649 RPM vs. 51,556.679 RPM, *p* < 10 × 10^−12^) and *miRNA-106a* (9.102 RPM vs. 8.111 RPM, *p* = 0.000077), as well as the down-regulation of *miRNA-143* (148,026.827 RPM vs. 452,590.255 RPM, *p* = 0.00000002) and *miRNA-551b* (0.691 RPM vs. 1.103 RPM, *p* = 0.000077). No differences were observed for *miRNA-122*, *miRNA-203a*, and *miRNA-574* (*p* > 0.05). Moreover, no differences (*p* > 0.05) were found in the case of *miRNA-122*, *miRNA-203a*, and *miRNA-574*, and all analyzed clinicopathological parameters were indicated. Expression levels of *miR-21* differed in the GC patients for I vs. III cancer stages (*p* = 0.0112), as well as between the adenocarcinoma diffuse and intestinal adenocarcinoma tubular (*p* = 0.0438). Surprisingly, *miRNA-106a* saw the most changes in expression levels depending on the patients’ race (Caucasian vs. Asian, *p* = 0.0160) and age (21–40 vs. 41–60, *p* = 0.0431; 21–40 vs. 61–80, *p* = 0.0081 and 61–80 vs. 81–100, *p* = 0.0172). Similarly, *miRNA-143* and *miRNA-203a* differ in the groups of patients between 41–60 vs. 61–80 years of age (*p* = 0.0092 and *p* = 0.0067, respectively). When we looked into tumor grade, only for patients with G2 vs. G3, differences in the expression levels for *miRNA-143* and *miRNA-203a* were noticed (*p* = 0.0018 and *p* = 0.0242, respectively). The expression levels of *miRNA-203a* were also associated with nodal metastasis status, and they differed depending on N0 vs. N3 (*p* = 0.0047), as well as N1 vs. N3 (*p* = 0.02911). For both *miRNA-143* and *miRNA-203a*, differences in expression levels were noted between the following: adenocarcinoma NOS vs. intestinal adenocarcinoma tubular (*p* = 0.000002 and *p* = 0.04314, respectively), adenocarcinoma diffuse vs. intestinal adenocarcinoma tubular (*p* = 0.00004 and *p* = 0.0056, respectively), intestinal adenocarcinomas NOS vs. tubular (*p* = 0.02458 and *p* = 0.02216, respectively), and intestinal adenocarcinomas mucinous vs. tubular (*p* = 0.00846 and *p* = 0.01260, respectively). Moreover, we indicated differences between adenocarcinoma NOS vs. intestinal adenocarcinoma mucinous (*p* = 0.02212) and intestinal adenocarcinomas mucinous vs. tubular (*p* = 0.00634) for *miRNA-106a*; between adenocarcinoma diffuse vs. intestinal adenocarcinoma papillary (*p* = 0.0082) for *miRNA-143*; and between adenocarcinomas NOS vs. signet ring (*p* = 0.00184), adenocarcinoma signet ring vs. intestinal adenocarcinoma NOS (*p* = 0.0052), as well as adenocarcinoma signet ring vs. intestinal adenocarcinoma tubular (*p* = 0.0001) for *miRNA-203a*. The last parameter, the *TP53* mutation status, was associated with differences in the expression levels of only *miRNA-143* (*p* = 0.006). All data are presented in Table 2 and in the UALCAN database website (https://ualcan.path.uab.edu/cgi-bin/TCGA-miR-HeatMap.pl?cancer=STAD, accessed on 7 June 2024).

### 2.2. Expression Levels of miR-143, miR-143*, and miR-574-3p before Surgery, as Well as miR-143* and miR-574-3p after, Were Up-Regulated in GC Patients

First of all, we checked the expression levels of the panel of miRNAs named *miRNA-21-3p*, *miRNA-21-5p*, *miRNA-106a-5p*, *miRNA-122-3p*, *miRNA-122-5p*, *miRNA-143-3p*, *miRNA-143-5p*, *miRNA-203a-3p*, *miRNA-203-5p*, *miRNA-551b-3p*, *miRNA-551b-5p*, and *miRNA-574-3p* before surgery, where biopsy was performed during gastroscopy, and after surgery, which was preceded by four cycles of neoadjuvant chemotherapy according to the FLOT or FLO regiment and in the normal samples taken from healthy individuals. Only in the case of *miRNA-143-5p*, *miRNA-143-3p*, and *miRNA-574-3p* did we observed significant down-regulation of those three miRNAs in patients before treatment in comparison to the normal samples taken from healthy individuals (*p* = 0.0117, *p* = 0.0434 and *p* = 0.0062, respectively). When we compared the changes in miRNA expression after treatment of the normal samples, significant changes were observed only in the case of *miRNA-143-3p* and *miRNA-574-3p* (*p* = 0.0325 and *p* = 0.0253, respectively). For the rest of the analyzed miRNAs, no differences (*p* > 0.05) were indicated. All results are presented in Figure 1 and Figure 2. The expression levels of all the determined miRNAs was not related to age, gender, or location of GC.

### 2.3. miR-143, miR-143*, and miR-574-3p Have Potential as Diagnostic Markers in GC Patients

Next, we checked if *miRNA-143-5p*, *miRNA-143-3p*, and *miRNA-574-3p* had potential as diagnostic markers, and the receiver operating characteristic curve (ROC) analyses of those three miRNAs in the patients’ samples taken before and after surgery were compared to the normal samples taken from the healthy, non-cancer individuals, and the area under the ROC curve (AUC) with 95% CI (confidence interval), as well as sensitivity and specificity, were calculated. We indicated *miRNA-143-5p* (AUC = 0.8500, 95% CI = 0.6459 to 1.000, *p* = 0.0129), *miRNA-143-3p* (AUC = 0.7875, 95% CI = 0.5404 to 1.000, *p* = 0.0410), and *miRNA-574-3p* (AUC = 0.9500, 95% CI = 0.8310 to 1.000, *p* = 0.0084), which displayed high ability as a potential diagnostic marker for distinguishing between patients before treatment in comparison to healthy patients, Figure 3.

### 2.4. miR-143 Could Be Used for Assessment of Therapy Response in GC Patients

In the analysis of the entire group of patients, comparing the miRNA level before and after adjuvant chemotherapy, we could demonstrate that *miR-143* could be a potential biomarker in response to chemotherapy, although it should be noted that not all cases showed histopathological and clinical improvement. A detailed analysis of these data is provided in the points below. *miR-143* showed a reduced expression level compared to the material after surgery, i.e., after chemotherapy. However, these data did not show statistical significance. We can notice a similar relationship in *miR-574-3p*. However, after additional analysis using the receiver operating characteristic curves (ROC) of *miR-21*, *miR-143*, and *miR-574-3p* in patients’ samples taken before and after surgery alongside box and whisker plots with a 5–95 percentile, the Wilcoxon matched-pairs signed-rank test showed that *miR-143* may be a potential biomarker of response to chemotherapy in LAGC (Figure 4). The other miRNA expressions were far from statistical significance in this analysis.

### 2.5. miR-551b* Could Be Used for Assessment of Therapy Response in GC Patients—Better than CT Scan?

We also assessed the miRNA expression in each patient and assessed the correlation with response to NAC according to the FLOT or FLO regimen. Responses to NAC were taken into account, such as histopathological (HP) scans and imaging, which were assessed based on computed tomography (CT). The expression of *miR-143-3p*, *miR-143-5p*, *miR-203a-3p*, and *miR-551b-5p* were found to be statistically significant in individual patients with a response to NAC (assessed by CT or/and HP (*p* > 0.05)). One patient had a *miR-143-3p* expression with a CT and HP response to NAC; three patients had *miR-143-5p* and *miR-203a-3p* expression with a CT and HP response to NAC. Additionally, *miR-551b-5p* expression was demonstrated in four cases; three cases had a CT and HP response to NAC, but in one case, as detected with a CT scan, the disease progressed, although the postoperative material showed high-grade regression in this patient. No expressions were observed for *miR-21-3p*, *miR-21-5p*, *miR-106a-5p*, *miR-122-3p*, *miR-122-5p*, *miR-203a-5p*, *miRNA-551b-3p*, and *miR-574-3p* in all cases (*p* > 0.05) (Table 3).

## 3. Discussion

The relationship between miRNA expression and the sensitivity of GC to chemotherapeutic drugs was studied extensively. We chose seven GC-related miRNAs for our study, i.e., *miR-551b-3p*, *miR-122*, *miR-21*, *miR-106a*, *miR-143*, *miR-574-3p*, and *miR-203.* The selected miRNAs either had already been tested for chemoresistance to selected drugs but were different from our chemotherapy regimens, or they were associated with the development of metastasis in GC. The miRNAs associated with metastasis may also play a major role in chemoresistance as both develop neoplasms. Moreover, the miRNAs previously tested for chemoresistance to other drugs may also play a role in the FLOT and FLO regimens. The mechanisms of chemotherapy resistance in GC, including the modulation of the miRNAs analyzed in this study, are presented in Figure 5. *MicroRNA-21* (*miR-21*) is one of the most frequently studied miRNAs. It has been proven that phosphatase and the tensin homolog is a direct target of *miR-21*, the expression of which is elevated in GC tissues and GC-derived cell lines [29]. Chan et al. demonstrated that *miR-21* was overexpressed in the GC tissues of 92% of patients compared to their non-GC counterparts. In addition, the overexpression of *miR-21* is associated with worse tumor differentiation, lymph node metastasis, and TNM stage [30]. Yang et al. reported that *miR-21* levels were increased in cisplatin-resistant GC cells (SCG7901/DDP) compared with the SGC7901 control group. Increased *miR-21* expression reduced the antiproliferative effect of apoptosisinduced by cisplatin. *miR-21* induces cisplatin resistance by directly down-regulating PTEN and activating the PI3K/AKT pathway [31]. Subsequent studies have assessed the involvement of *miR-21* in the development of resistance to paclitaxel (PTX) in GC cells. In addition, increased *miR-21* expression levels have been demonstrated in PTX-resistant SGC7901 cells (SGC7901/PTX) compared to parental SGC7901 cells. The increased expression of *miR-21* in SGC7901 induces a decrease in antiproliferative activity and PTX-induced apoptosis. Next, the effect of *miR-21* through the regulation of P-gp was examined. It was found that treatment with *miR-21* mimics increased P-gp expression in wild-type SGC7901 cells. Treatment with *miR-21* inhibitors reduced the levels of the *ABCB1* mRNA and P-gp protein expression in SGC7901/PTX cells. These data suggest that *miR-21* may act by regulating P-gp expression, which is involved in the PTX resistance in SGC7901/PTX cell lines [32]. The TCGA database showed the up-regulation of *miR-21-3p* and *miR-21-5p* in patients with GC. Moreover, the UALCAN database showed that the expression of *miR-21* differs in GC patients for I vs. III cancer stages. Differences in the expression at certain stages may be helpful in the correct staging of patients with LAGC if computed tomography (CT) is ambiguous. Computed tomography is a standard examination technique used to assess the clinical stage of gastric cancer. Based on CT, the patient is also qualified for surgical treatment after neoadjuvant chemotherapy. However, recent research indicates that this test has low sensitivity and overestimates the degree of clinical advancement. This may result in the patient being disqualified from radical LAGC removal surgery. Upstaging may be related to the incorrect interpretation of fibrosis after NAC or inflammation in CT. Therefore, it is very important to find an additional tool that will allow for a more precise assessment of the clinical stage after NAC in patients with LAGC [33]. Our research was not consistent with the research results described above. We have not demonstrated the expression of *miR-21-3p* and *miR-21-5p* in LAGC in either preoperative or postoperative material. The discrepancy between the results of our study and those of others may be related to the material examined or the small number of cases. We used FFPET in our studies, and the studies cited above used cell lines. Due to the inconsistency, studies involving this miRNA should be continued to clearly indicate its role in the development of GC. The next detected miRNA was *miR-106a-5p*. The overexpression of the tumor suppressor lncRNA *GAS5* inhibits GC development through percolating *miR-106a-5p* through the *Akt/mTOR* pathway. Decreased levels of *GAS5* and increased levels of *miR*-106a-5p were demonstrated in the cell lines and GCs. The *GAS5* level was significantly inversely correlated with the *miR-106a-5p* level in GC. Additionally, it has been shown that *GAS5* binds to *miR-106a-5p* and negatively regulates its expression in GC cells. *GAS5* overexpression inhibits GC cell proliferation, migration, and invasion, as well as promotes apoptosis. Moreover, the overexpression of *miRNA-106a-5p* reverses the functional effects induced by *GAS5* overexpression. In vivo, *GAS5* overexpression inhibited tumor growth by negatively regulating *miR-106a-5p* expression. In vitro and in vivo, *GAS5* overexpression inactivated the *Akt/mTOR* pathway by suppressing *miR-106a-5p* expression [34]. *miR-106a* showed higher expression levels in SGC7901/CDDP cells compared to parental SGC7901 cells. Additionally, transfection with a *miR-106a* mimetic induced CDDP resistance in wild-type SGC7901. The suppression of *miR-106a* in SGC7901/CDDP led to increased cisplatin cytotoxicity. The analysis showed that *PTEN* is a conserved target gene of *miR-106a*. There was a strong inverse correlation between *miR-106a* and *PTEN* levels. The overexpression of *miR-106a* activated the *PI3K/AKT* pathway through its inhibitory role on *PTEN* [35]. Other studies have shown that the hypermethylation of *TFAP2E* results in its reduced expression and chemoresistance to 5-FU in GC cells. The strong expression of *miR-106a-5p* and *miR-421* regulated the chemoresistance induced by *TFAP2E* methylation [36]. The TCGA database also showed the overexpression of *miR-106a* in GC compared to normal tissue. Moreover, we checked the UALCAN database, and *miR-106a* exhibited the most changes in the expression levels depending on the patients’ race and age. In our results, we did not demonstrate a statistically significant relationship between the expression of *miR-106a* and the development of chemoresistance in GC. The discrepancy in the data may be related to the different material used and the action of *miR-106a.* The cited studies indicate that *miR-106a* influences the development of GC by taking part in complex cellular pathways. Perhaps we should follow this lead in research to determine the roles of *miR-106a* in GC. However, it is worth considering whether *miR-106a* will be an appropriate tool for the diagnosis of GC or response to NAC in patients with LAGC. The high variability in the expression levels according to age and race excludes this parameter as a potential biomarker. Recent studies have shown that *miR-122* is also overexpressed in GCs and contributes to tumor growth and drug resistance. *MicroRNA-122* (*miR-122*) acts as a tumor suppressor in various cancers, including GC. In their study, Meng et al. demonstrated a low level of *miR-122-5p* expression in GC tissues and cell lines. Additionally, by targeting *LYN*, *miR-122-5p* overexpression inhibited GC cell proliferation, migration, and invasion. The expression of *LYN*, an Src family tyrosine kinase, was inversely correlated with the expression of *miR-122-5p* in GC tissues [37]. Further, decreased *miR-122* expression is directly involved in the induction of cisplatin (CDDP) resistance by increasing the excision repair cross-complementing 1 (*ERCC1*) expression [38]. The TCGA database does not show the *miR-122* expression in GC, which was confirmed by our research. This means that our results are not consistent with the cited literature. Interestingly, the abovementioned studies were also conducted on tissue material from GC. The discrepancies may result from the type of tool or materials used for testing. The inconsistency of our results, the literature data, and the TCGA database clearly requires further research. The overexpression of *miR-143* has a negative effect on MKN-45 cell proliferation and invasion. Additionally, the downstream targets of *miR-143* were assessed: the GC cells showed reduced expressions of *K-Ras*, *MMP9*, and *C-Myc*, as well as increased expressions of *Bax*, *caspase-3*, and *caspase-9* [39]. Data from TCGA also showed decreased *miR-143-3p* expression in GC compared to normal tissue. *miR-143-3p* was detected in both tumor tissue and plasma, which makes it an even more interesting potential biomarker for GC detection. *miR-143* is involved in the development of cisplatin resistance via *IGF1R* and *BCL2*. *miR-143* expression is decreased in human GC cell lines and in the cisplatin-resistant GC cell line SGC7901/cisplatin (DDP). It is also related to an increase in the levels of *IGF1R* and *BCL2* compared to the parent SGC7901 cell line. It is suggested that they are target genes of *miR-143*. The overexpression of *miR-143* sensitizes SGC7901/DDP cells to cisplatin-induced apoptosis and inhibits proliferation [40]. *miR-143*, a potent inhibitor of autophagy, enhances the chemosensitivity to quercetin through autophagy inhibition via the target GABARAPL1 in GC cells [41]. Decreased *miR-143-3p* expression correlates with late-stage and lymph node metastasis. *miR-143-3p* also negatively regulates cell growth, apoptosis, migration, and invasion by directly targeting the *AKT2* gene [42]. The TCGA database demonstrated a decreased expression of *miR-143-5p* in GC patients. Our studies confirm the involvement of *miR-143* in the development of GC. Our study showed a decreased expression of *miR-143-3p* and *miR-143-5p* preoperatively and *miR-143-3p* postoperatively after NAC in LAGC patients. Additional analyses showed that *miR-143-3p* and *miR-143-5p* may be potential tools for detecting GC. The consistency of our studies with the literature and the TCGA database indicates a significant role of *miR-143* in chemoresistance in GC. Further studies are needed to more specifically identify the roles of *miR-143-3p* and *miR-143-5p* in the development of GC in order to improve future clinical practice. Many studies show that *miR-203a* inhibits invasion, growth, and metastasis by regulating multiple pathways in GC. Studies indicate that *miR-203a-3p* is decreased in both GC tissues and cell lines. Moreover, the overexpression of *miR-203a-3p* reduced GC cell proliferation and cell cycle progression in vitro. In GC cells, *miR-203a-3p* can inhibit tumor development by negatively regulating *IGF-1R* expression. In GC cells, insulin-like growth factor 1 receptor (*IGF-1R*) is a target mediator of *miR-203a-3p* [43]. Other studies have shown that *miR-203a* expression is decreased in GC. Moreover, *miR-203* expression was associated with the radiosensitivity of GC cells because it promoted cell apoptosis in GCs subjected to radiotherapy by targeting the Zinc finger E-box binding homeobox 1 (*ZEB1*) [44]. The TCGA database did not show numerical data on *miR-203a-3p* and *miR-203-5p* in GC. Only graphic forms were revealed. Our research does not confirm the data from the studies cited above. Our studies have not demonstrated the expression of *miR-203a* in patients with GC. However, a detailed analysis showed *miR-203a* expression in three patients responding to NAC according to the FLOT or FLO regimen. These data should encourage further research to assess the role of *miR-203a* in assessing the response to NAC in LAGC. The next miRNA of note in assessing chemoresistance in GC was *miR-551b*. Guo et al. demonstrated that the lncRNA-*GC1*–*miRNA-551b-3p*–dysbindin signaling pathway can serve as a predictor of the response to oxaliplatin. lncRNA-*GC1* and *miRNA-551b-3p* were elevated in chemotherapy-resistant GC. *miR-551b-3p* binds to the non-coding region of dysbindin mRNA, thereby negatively regulating dysbindin expression, which is involved in chemoresistance in GC cells. Additionally, it has been shown that IncRNA-*GC1* increases chemoresistance in GC through competitive binding with *miR-551b-3p* [45]. A recent study reported that *miR-551b-3p* directly binds to the intronic region of dysbindin mRNA, negatively regulating its expression, and it is involved in platinum resistance in GC cells [38]. The analyzed TCGA database showed an increased level of *miR-551b-3p* in GC patients compared to normal tissue. Interestingly, we did not demonstrate the overexpression of *miR-551b-3p* or *miR-551-5p* in GC patients. The statistical data do demonstrate a potential role of *miR-551b* in the response to NAC in GC, or as a potential biomarker in the detection of GC. However, after analyzing the individual cases, *miR-551b-5p* showed increased levels of expression in four patients. Additionally, two patients showed discordant responses to NAC in LAGC on CT and HP. The CT scan showed a higher stage of GC than in the HP examination, and in one case, CT showed progression and the HP examination showed a complete regression of GC in the same case. The UALCAN database does not show statistically significant differences in the expression of *miRNA-551b-5p* between clinical stages of GC. However, the staging of GC described by the UALCAN database was only clinical. Our study was more detailed and drew attention to the problem of inconsistency between the clinical stage and pathological stage in GC after NAC. Therefore, the results may be different. Our data are innovative, although we need larger case studies to evaluate the role of *miR-551b-3p* and the role of *miR-551b-5p* in response to NAC in LAGC. The last interesting miRNAs involved in chemoresistance in gastric cancer were *miR-574-3p* and *miR-574-5p*. *miR-574-5p* is involved in GC by promoting angiogenesis [46]. Under hypoxic conditions, the expression of *miR-574-5p* increases. The inhibition of *miR-574-5p* reduces the expression of endothelial growth factor A (VEGFA). In gastric cancer cells, *miR-574-5p* promotes angiogenesis by increasing p44/42 MAPK phosphorylation through the inhibition of *PTPN3* expression [46,47]. Furthermore, Wang et al. showed that the overexpression of *miR-574-3p* reduces the migratory and invasive properties of the GC cells, which inhibits the EMT, as well as enhances cisplatin sensitivity in the GC cells in vitro and in vivo through suppression [48]. Additionally, a study has shown that the reduced expression of *miR-574-3p* occurs mainly in the early stages of GC or in cancers with a high level of differentiation, suggesting that it may be used as a marker for mild cases of GC [49]. Zhiwu et al. showed that *miR-574-3p* was overexpressed in GC tissues and cells. *miR-574-3p* targets Cullin 2 (*CUL2*), increasing *HIF-1α* expression and affecting GC progression [50]. The TCGA database has not shown increased *miR-574* values in GC. However, the results of our research coincide with those of the studies cited above. In our studies, we showed statistically significantly decreased *miR-574-3p* expression before and after surgery in patients with GC. Additional analyses demonstrated the potential role of *miR-574-3p* as a tool for GC diagnosis. Our study has several limitations, including the material used, FFPET, and the small number of cases. FFPET samples present limitations for RNA-based molecular studies due to the RNA degradation caused by the formalin fixation and embedding process, which results in a marked reduction in detectable mRNA molecules. This process causes the enzymatic degradation and chemical modification of RNA, giving rise to cross-links with proteins and making RNA extraction difficult. Thus, a digestion step with proteinase K is required to eliminate cross-links and facilitate RNA extraction from FFPET samples. The longer the RNA molecule, the more likely cross-links will remain after proteinase K digestion. Hence, small RNA molecules will be easier to extract from this type of sample, and fragments larger than 200 nt will be harder to recover. Another limitation of our study is the number of cases and controls examined. However, these are preliminary studies that we plan to expand upon to verify our results.

## 4. Materials and Methods

### 4.1. Patient Criteria Included in the Study and Sample Preparations

We collected 98 histologically confirmed LAGC patients between January 2018 and December 2022. The patients had to meet the inclusion criteria described below and not meet the exclusion criteria, which is also described below. Finally, 10 patients with LAGC were included in the analysis. We required two histological materials from each selected LAGC patient. The first histological material came from a biopsy taken during gastroscopy. This was the material on which the diagnosis of LAGC was made. Then, these patients were qualified for perioperative chemotherapy. These patients received 4 cycles of neoadjuvant chemotherapy according to the FLOT or FLO regimen. Further histological material from GC was obtained from these patients after radical gastrectomy. The eligibility criteria were as follows: preoperative cT2–4, histologically proven adenocarcinoma, neoadjuvant chemotherapy according to the FLOT or FLO regimen, complete clinical records, and no distant metastasis such as in the liver, lung, or bone. The exclusion criteria were as follows: having a previous history of other cancers and/or had received preoperative radiotherapy. The patients that qualified for neoadjuvant chemotherapy received either the FLOT or FLO regimen every 2 weeks depending on their clinical condition. The FLOT regimen consisted of docetaxel (60 mg/m^2^), oxaliplatin (85 mg/m^2^), leucovorin (200 mg/m^2^), and 5-fluorouracil (2.600 mg/m^2^ as a 24 h infusion), all given on Day 1, and the FLO regimen was conducted without docetaxel. Patients received 4 cycles prior to elective surgery. Patients underwent imaging evaluation CT after neoadjuvant chemotherapy. If the tumor size decreased or was stable, the operation was performed at the earliest available time in the Department of Oncological Surgery. Responses to chemotherapy were evaluated by CT scan after four cycles according to RECIST criterion v 1.1, and these were then compared with the baseline CT scan performed before treatment. Additional responses to chemotherapy were evaluated in histological postoperative material. All classification is presented in Table 4. The control group was composed of 7 tissue materials from the gastric area without cancer. The material was obtained during gastrectomy. Studies were carried out on a group of 6 men and 4 women with GC. The patients’ ages ranged from 40 to 77 years, and the mean age of the patients was 61 years (Table 5).

### 4.2. Ethical Issues

This study was carried out with the approval of the local ethics committee and is based on archival material—formalin-fixed paraffin-embedded tissue (FFPET) section blocks from surgical specimens. All analyses included in this study do not bear any traces of a medical experiment in accordance with the law in the Republic of Poland (the ethics committee of the Poznan University of Medical Sciences, No. KB 391/24).

### 4.3. Sample Preparation

All samples were clinically and histologically confirmed by pathologists based on tumor tests that were performed on the formalin-fixed paraffin-embedded tissue (FFPET) section blocks from surgical specimens using hematoxylin and eosin (H&E) histological stains, and they were rated by microscopic observation. Next, cancer cells were marked on the FFPET section blocks and sliced into pieces about 10 μm in thickness for RNA isolation.

### 4.4. Total RNA Isolation

The total RNA from the FFPET slides were isolated using a GeneMATRIX FFPE RNA Purification Kit (EURx Sp. z o.o., Gdansk, Poland) in accordance with the manufacturers’ protocol. Briefly, one FFPET tissue preparation was approximately 10 μm thick, and the formaldehyde/paraffin was removed by dissolving and removing the paraffin using the xylene/heptane/methanol method. After removing the supernatant, the pellet was allowed to dry and the RNA isolation procedure began. Dry tissue pellets were suspended in a Lyse ALL solution, mixed, and Proteinase K was added, the pellets were then incubated at 56 °C and then at 80 °C. The samples were cooled and centrifuged at maximum speed. The obtained supernatants were transferred to a new tube and incubated with RL buffer, and 96–100% ethyl alcohol was next added and mixed. All were transferred into the homogenization mini columns and centrifuged. Next, the columns were washed using Wash RNA buffer and centrifugation. To the obtained supernatant, DNRII buffer and DNase I were added, which was then incubated; after that, RL buffer and 96–100% ethyl alcohol were added and mixed. Prepared mixes were transferred to the RNA-binding mini-column, then washed twice by Wash buffer and centrifugation. Completely dry spin columns were placed in new Eppendorf tubes and RNase-free water, and centrifugation was then applied to release RNA molecules. 

Next, the quality and quantity of the isolated RNA samples were examined using the NanoDrop 2000 spectrophotometer (Thermo Scientific, Waltham, MA, USA). After that, the RNA was stored at −80 °C until used.

### 4.5. Assessment of miRNA Expression Levels

We used a preselected panel of 12 miRNAs, composed of *miRNA-21-3p* (MIMAT0004494, assay ID: 002438); *miRNA-21-5p* (MIMAT0000076, assay ID: 000397); *miRNA-106a-5p* (MIMAT0000103, assay ID: 002169); *miRNA-122-3p* (MIMAT0004590, assay ID: 002130); *miRNA-122-5p* (MIMAT0000421, assay ID: 002245); *miRNA-143-3p* (MIMAT0000435, assay ID: 002249); *miRNA-143-5p* (MIMAT0000435, assay ID: 002249); *miRNA-203a-3p* (MIMAT0000264, assay ID: 000507); *miRNA-203a-5p* (MIMAT0031890, assay ID: 477013_mat); *miRNA-551b-3p* (MIMAT0003233, assay ID: 001535); *miRNA-551b-5p* (MIMAT0004794, assay ID: 002346); and *miRNA-574-3p* (MIMAT0003239, assay ID: 002349), as well as *U6* snRNA (NCBI Accession: NR_004394, assay ID: 001973) as a reference gene, which are commercially available primers that were obtained from TaqMan™ MicroRNA Assay (Catalog number: 4427975, Applied Biosystems, Foster City, CA, USA). The miRNA expression levels were defined by a two-step qRT-PCR method using a TaqMan microRNA Assay (Applied Biosystems, Foster City, CA, USA) in accordance with the manufacturer’s protocol and using a LightCycler 96 thermocycler (Roche, Basel, Switzerland) as described previously [51].

### 4.6. miRNA Calculation

The miRNA expressions were analyzed by quantitative reverse transcriptase polymerase chain reaction (qRT-PCR). Profiles of miRNA were prepared using preoperative biopsies without prior therapy, and the next were prepared after neoadjuvant therapy (i.e., we used the material after surgery for the LAGC patients). Obtained cycle threshold (CT) values were calculated using the 2^−ΔCT^ method and through normalizing against the mean of U6 snRNA expression for each sample as described previously [52]. The chosen TaqMan microRNA Assay enables determinations of the level of mature forms of miRNA and their differentiation, with an accuracy of one nucleotide in the sequence of tested miRNAs, with high accuracy and sensitivity [53].

### 4.7. Databases

For assessment of the expression levels of *miRNA-21-3p* (MIMAT0004494), *miRNA-21-5p* (MIMAT0000076), *miRNA-106a-5p* (MIMAT0000103), *miRNA-122-3p* (MIMAT0004590), *miRNA-122-5p* (MIMAT0000421), *miRNA-143-3p* (MIMAT0000435), *miRNA-143-5p* (MIMAT0004599), *miRNA-203a-3p* (*miRNA-203-3p;* MIMAT0000264), *miRNA-203-5p* (), *miRNA-551b-3p* (MIMAT0003233), *miRNA-551b-5p* (MIMAT0004794), and *miRNA-574-3p* (MIMAT0003239) in stomach adenocarcinoma (STAD) patients and in normal samples, we used the ENCORI database presented as log2 [RPM + 0.01] [32] (https://rnasysu.com/encori/panMirDiffExp.php, accessed on 7 June 2024), which was included in the TCGA project. Moreover, differences in the miRNA expression levels (RPM, reads per million) depending on clinicopathological parameters, including sample type, race, gender, age, cancer stage, tumor grade, nodal metastasis status, tumor histology, and the *TP53* mutation status of STAD patients, were taken and analyzed from the UALCAN database (accessed on 7 June 2024). Only the data of miRNAs named there as *miRNA-21*, *miRNA-106a*, *miRNA-122*, *miRNA-143*, *miRNA-203a*, *miRNA-551b*, and *miRNA-574* were available [54].

The data used and presented in this study are openly available at the TCGA-based databases and do not violate any copyrights.

### 4.8. Statistical Analysis

We used GraphPad Prism9 (GraphPad, San Diego, CA, USA) for calculation of all statistical analyses. The *t*-test and Mann–Whitney U test were used depending on the data normality estimated using the Shapiro–Wilk normality test. All *t*-tests and ANOVA tests were performed as two-tailed and considered significant at *p* < 0.05, similar to that described previously [55].

## 5. Conclusions

The *miR-143-3p*, *miR-143-5p*, *miR-203a-3p*, and *miR-551b-5p* expression in paraffin blocks with tissue material can help differentiate between responders and non-responders to NAC in LAGC. In our research, we used tissue material in the form of paraffin blocks obtained from biopsies of LAGC patients before surgery, as well as material obtained after surgery. This could allow for the non-invasive assessment of chemoresistance in LAGC patients. Moreover, *miR-143-3p*, *miR-143-5p*, and *miR-574-3p* may be used as diagnostic tools in GC patients. *miR-551b-5p* may support the correct assessment of the response to NAC in GC by CT. Our preliminary research is innovative, and the results are promising. The study has many limitations, but our research results should indicate the direction for further research.

## Figures and Tables

**Figure 1 ijms-25-08057-f001:**
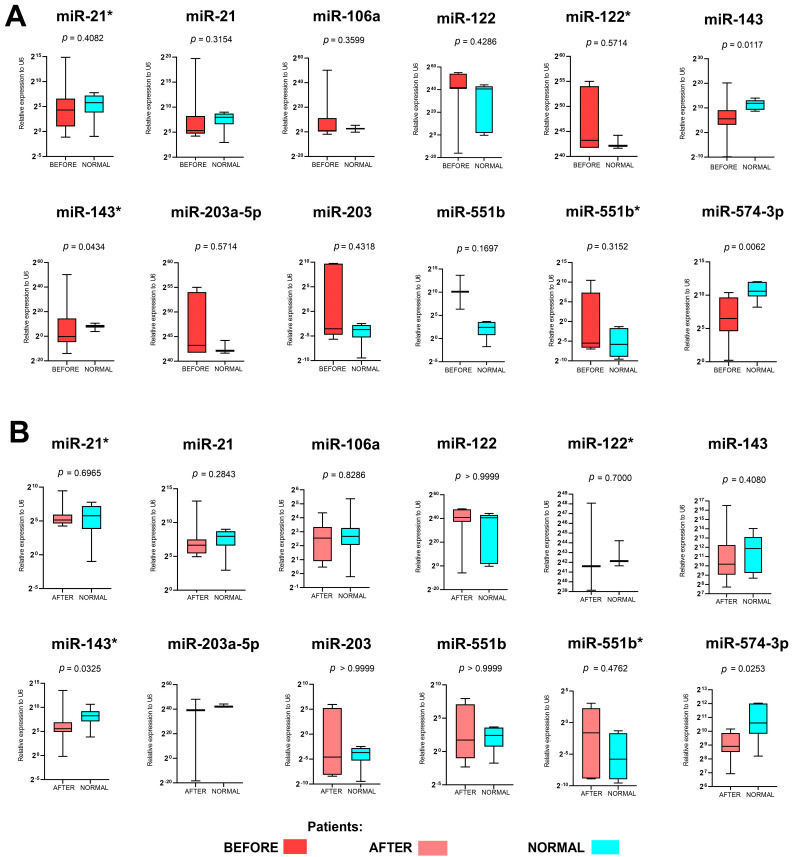
Expression levels of *miR-21**, *miR-21*, *miR-106a*, *miR-122*, *miR-122**, *miR-143*, *miR-143**, *miR-203a-5p*, *miR-203*, *miR-551b*, *miR-551b**, and *miR-574-3p* before surgery (**A**), where biopsy was performed during gastroscopy, and after surgery (**B**), which was preceded by four cycles of neoadjuvant chemotherapy according to the FLOT or FLO regiment and in the normal samples taken from healthy individuals. Box and whisker plots with a 5–95 percentile and Mann–Whitney test. A *p* of <0.05 was considered significant.

**Figure 2 ijms-25-08057-f002:**
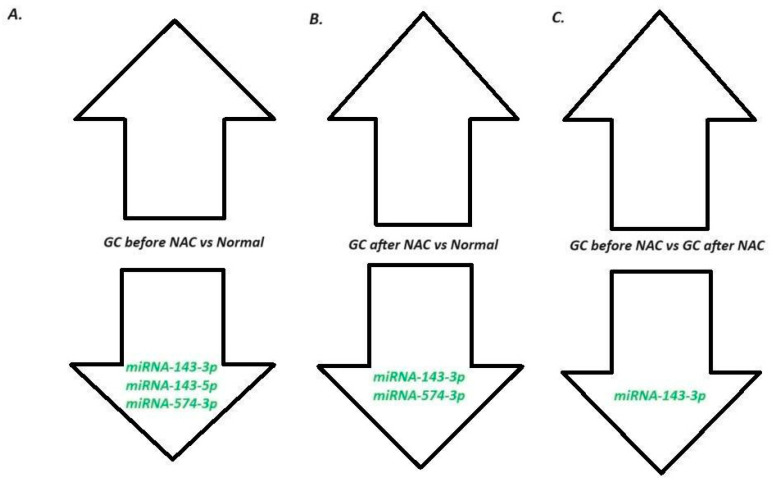
A schematic diagram of our results: (**A**) Comparison of the miRNA expressions in LAGC before NAC (pre-surgery) and in normal tissue. (**B**) Comparison of the miRNA expression in LAGC after NAC (post-surgery) and in normal tissue (**C**) Comparison of the miRNA expression pre- (before NAC) and post-surgery (after NAC). miRNA—microRNA; GC—gastric cancer; and NAC—neoadjuvant chemotherapy.

**Figure 3 ijms-25-08057-f003:**
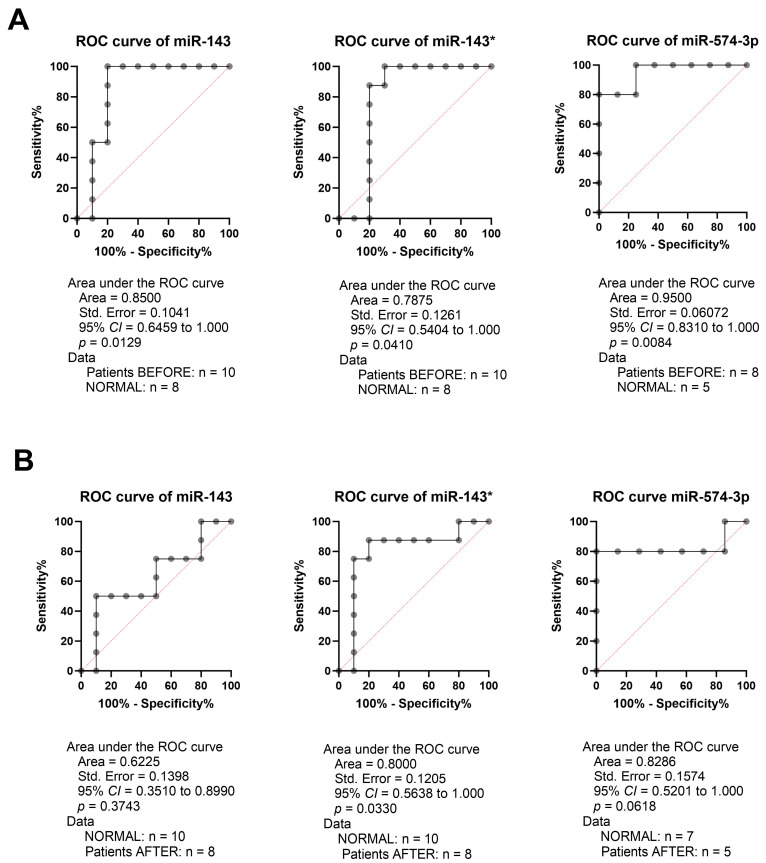
Receiver operating characteristic curve (ROC) analyses of *miR-143*, *miR-143**, and *miR-574-3p* in patients’ samples taken (**A**) before and (**B**) after surgery in comparison to the normal samples taken from healthy, non-cancer individuals. CI—confidence interval and n—number of cases in analyses. A *p* of <0.05 was considered significant.

**Figure 4 ijms-25-08057-f004:**
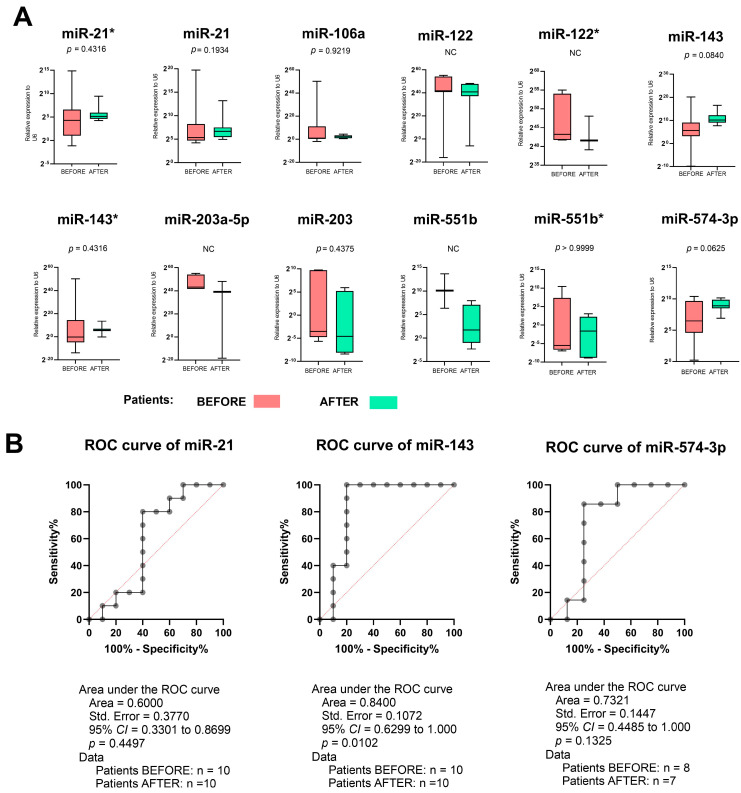
Expression levels of *miR-21**, *miR-21*, *miR-106a*, *miR-122*, *miR-122**, *miR-143*, *miR-143**, *miR-203a-5p*, *miR-203*, *miR-551b*, *miR-551b**, and *miR-574-3p* (**A**) before surgery, where a biopsy was performed during gastroscopy, and after surgery, which was preceded by four cycles of neoadjuvant chemotherapy according to the FLOT or FLO regiment. (**B**) The receiver operating characteristic curve (ROC) analyses of *miR-21*, *miR-143*, and *miR-574-3p* in the patient samples taken before and after surgery. Box and whisker plots with a 5–95 percentile and the Wilcoxon matched-pairs signed-rank test. CI—confidence interval, n—number of cases in analyses, NC—not calculated due to a lack of gene expression. A *p* of <0.05 was considered significant.

**Figure 5 ijms-25-08057-f005:**
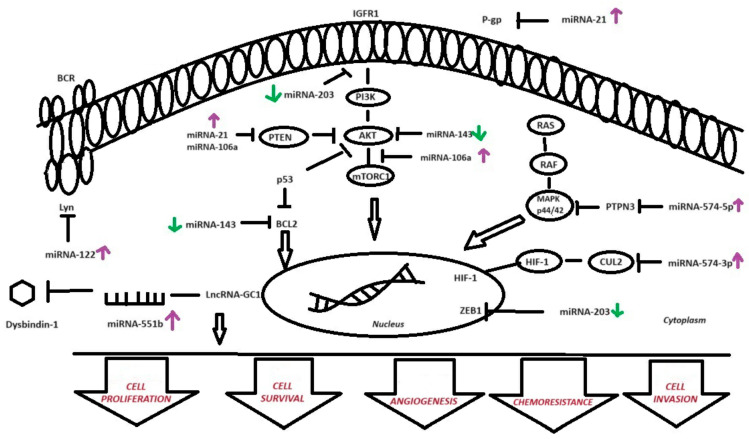
Schematic diagram of drug resistance development induced by *miRNA-21*, *miRNA-106a-5p*, *miRNA-122-5p*, *miRNA-143*, *miRNA-203*, *miRNA-551b-3p*, *miRNA-574-3p*, and *miRNA-574-5p* in GC cells [31,32,35,40,42,44,45,46,47,50]. *miR-21* induces cisplatin resistance by directly down-regulating PTEN and activating the *PI3K/AKT* pathway. Moreover, *miR-21* may regulate *P-gp* expression, which is involved in paclitaxel resistance in GC. The *Akt/mTOR* pathway is suppressed by *miRNA-106a-5p* expression. Overexpression of *miR-106a* activates the PI3K/AKT pathway through its inhibitory effect on *PTEN*. The suppression of *miR-106a* in GC leads to increased cisplatin cytotoxicity. Furthermore, by targeting *LYN*, *miR-122-5p* overexpression inhibits GC cell proliferation, migration, and invasion. *miR-143-3p* negatively regulates cell growth, apoptosis, migration, and invasion by directly targeting the *AKT2* gene. *miR-203* expression is also associated with the radiosensitivity of GC cells because it promotes cell apoptosis in GCs by targeting *ZEB1*. lncRNA-*GC1* and *miRNA-551b-3p* were elevated in oxaliplatin-resistant GC. *miR-551b-3p* binds to the noncoding region of dysbindin mRNA, thereby negatively regulating dysbindin expression, which is involved in the chemoresistance in GC cells. Under hypoxic conditions, the expression of *miR-574-5p* increases. In GC cells, *miR-574-5p* promotes angiogenesis by increasing p44/42 MAPK phosphorylation through the inhibition of *PTPN3* expression. *miR-574-3p* targets CUL2, increasing HIF-1α expression and affecting GC progression. miRNA: microRNA; GC: gastric cancer; *ZEB1*: zinc finger E-box binding homeobox 1; and *CUL2*: Cullin 2; green—over-expressed miRNAs, and purple—under-expressed miRNAs.

**Table 1 ijms-25-08057-t001:** Expression levels of the selected miRNAs in the cancer and normal samples from gastric patients analyzed during the TCGA project. Data taken from the ENCORI database. n—number of samples, FDR—false discovery rate, green—over-expressed miRNAs, and purple—under-expressed miRNAs.

miRNA	Cancer Samples [n]	Normal Samples [n]	Median Expression Level in Cancer Samples	Median Expression Level in Normal Samples	Fold Change	*p*-Value	FDR
** *miRNA-21-3p* **	372	32	3285.16	1165.75	2.82	**6.8 × 10^−24^**	2.5 × 10^−21^
** *miRNA-21-5p* **	372	32	282,109.88	64,062.21	4.4	**1.83 × 10^−68^**	4.7 × 10^−65^
** *miRNA-106a-5p* **	372	32	18.72	7.85	2.38	**0.00099**	0.0068
** *miRNA-122-3p* **	372	32	0.21	0.01	20.55	0.25	0.6
** *miRNA-122-5p* **	372	32	17.3	0.57	29.92	0.54	0.83
** *miRNA-143-3p* **	372	32	193,518.49	443,164.16	0.44	3.6 × 10^−10^	9.3 × 10^−9^
** *miRNA-143-5p* **	372	32	121.59	229.49	0.53	**0.011**	0.054
** *miRNA-203a-3p* **	372	32	-	-	-	-	-
** *miRNA-203a-5p* **	372	32	-	-	-	-	-
** *miRNA-551b-3p* **	372	32	1.53	07.02	0.22	0.037	0.15
** *miRNA-551b-5p* **	372	32	0.02	0.03	0.56	0.13	0.39
** *miRNA-574-3p* **	372	32	81.06	85.97	0.94	0.2	0.54

**Table 2 ijms-25-08057-t002:** Differences in the expression levels of *miRNA-21*, *miRNA-106a*, *miRNA-122*, *miRNA-143*, *miRNA-203a*, *miRNA-551b*, and *miRNA-574* depending on the clinicopathological parameters in GC patients based on the TCGA project. Data taken from the UALCAN database. AC—adenocarcinoma, IAC—intestinal adenocarcinoma, green—over-expressed miRNAs, and purple—under-expressed miRNAs. A *p* of <0.05 was considered significant and is marked in orange.

Parameter	Groups	*miRNA-21*	*miRNA-106a*	*miRNA-122*	*miRNA-143*	*miRNA-203a*	*miRNA-551b*	*miRNA-574*
Sample type	Normal vs. Primary	**<1 × 10^−12^**	**0.00008**	0.14658	**0.00000**	0.16578	**0.00242**	0.69254
Race	Caucasian vs. African American	0.59606	0.65550	0.19030	0.05878	0.66776	0.05402	0.05402
	Caucasian vs. Asian	0.30630	**0.01604**	0.17484	0.02172	0.31662	0.63522	0.63522
	African American vs. Asian	0.29902	0.56788	0.62328	0.29880	0.99920	0.13424	0.13424
Gender	Male vs. Female	0.74084	0.53014	0.24308	0.32242	0.28556	0.56872	0.56872
Age	21–40 vs. 41–60	0.14104	0.04318	0.18056	0.43436	0.22830	0.41534	0.41534
	21–40 vs. 61–80	0.25226	**0.00808**	0.12376	0.76562	0.77240	0.05157	0.05157
	21–40 vs. 81–100	0.05878	0.34576	0.90334	0.55958	0.13628	0.08584	0.08584
	41–60 vs. 61–80	0.08270	0.23986	0.20184	**0.00918**	0.00672	0.93720	0.93720
	41–60 vs. 81–100	0.49906	0.15064	0.17968	0.56066	0.61330	0.55180	0.55180
	61–80 vs. 81–100	0.11083	**0.01720**	0.10224	0.43432	0.06366	0.63018	0.63018
Cancer stage	I vs. II	0.15316	0.16144	0.29706	0.47704	0.43726	0.54456	0.54456
	I vs. III	**0.01124**	0.60368	0.32454	0.54468	0.05878	0.56204	0.56204
	I vs. IV	0.23102	0.53414	0.36276	0.40824	0.07544	0.75392	0.75392
	II vs. III	0.15882	0.23550	0.47678	0.87786	0.12802	0.90778	0.90778
	II vs. IV	0.88272	0.49128	0.25600	0.68498	0.15774	0.28022	0.28022
	III vs. IV	0.48102	0.81706	0.58708	0.62610	0.89714	0.35976	0.35976
Tumor grade	G1 vs. G2	0.67776	0.74114	0.30394	0.79310	0.60826	0.22676	0.22676
	G1 vs. G3	0.65476	0.90388	0.09204	0.57692	0.17936	0.13166	0.13166
	G2 vs. G3	0.86524	0.34396	0.46276	**0.00182**	**0.02424**	0.83656	0.83656
Nodal metastasis status	N0 vs. N1	0.30396	0.88494	0.27724	0.78988	0.75946	0.83018	0.83018
	N0 vs. N2	0.29180	0.71396	0.29278	0.71558	0.80560	0.72304	0.72304
	N0 vs. N3	0.71348	0.49888	0.42276	0.28858	**0.00474**	0.17108	0.17108
	N1 vs. N2	0.96318	0.88320	0.55314	0.56090	0.98958	0.89646	0.89646
	N1 vs. N3	0.78694	0.48110	0.35356	0.24474	**0.02912**	0.15342	0.15342
	N2 vs. N3	0.75010	0.39538	0.42578	0.41616	0.09393	0.14404	0.14404
Tumor histology	AC NOS vs. AC Diffuse	0.52732	0.75522	0.22130	0.21224	0.20154	0.13558	0.13558
	AC NOS vs. AC Signet Ring	0.12324	0.13950	0.65222	0.62216	**0.00184**	0.10358	0.10358
	AC NOS vs. IAC NOS	0.45426	0.55860	0.36642	0.11122	0.66980	0.92958	0.92958
	AC NOS vs. IAC Mucinous	0.39568	**0.02212**	0.28880	0.43058	0.33528	0.12252	0.12252
	AC NOS vs. IAC Papillary	0.55152	0.84612	0.94758	0.09892	0.27432	0.47208	0.47208
	AC NOS vs. IAC Tubular	0.09716	0.55192	0.32894	**0.00000**	0.04314	0.09158	0.09158
	AC Diffuse vs. AC Signet Ring	0.07644	0.29140	0.33310	0.36696	0.06108	0.53796	0.53796
	AC Diffuse vs. IAC NOS	0.21652	0.46906	0.27970	**0.02094**	0.37026	0.14580	0.14580
	AC Diffuse vs. IAC Mucinous	0.66654	0.06158	0.30358	0.89524	0.93058	0.78986	0.78986
	AC Diffuse vs. IAC Papillary	0.40010	0.96520	0.16990	**0.00822**	0.19376	0.90458	0.90458
	AC Diffuse vs. IAC Tubular	**0.04388**	0.86194	0.31354	**0.00004**	0.00562	0.78714	0.78714
	AC Signet Ring vs. IAC NOS	0.22388	0.22494	0.52088	0.81776	0.00520	0.11222	0.11222
	AC Signet Ring vs. IAC Mucinous	0.10170	0.11510	0.35920	0.42808	0.09626	0.67820	0.67820
	AC Signet Ring vs. IAC Papillary	0.61504	0.45914	0.56958	0.37434	0.11228	0.62612	0.62612
	AC Signet Ring vs. IAC Tubular	0.42724	0.38018	0.37260	0.33454	**0.00010**	0.58822	0.58822
	IAC NOS vs. IAC Mucinous	0.19918	0.17518	0.28926	0.09030	0.56498	0.13216	0.13216
	IAC NOS vs. IAC Papillary	0.75650	0.64798	0.35850	0.29404	0.24576	0.63370	0.63370
	IAC NOS vs. IAC Tubular	0.41558	0.26374	0.29396	**0.02458**	**0.02216**	0.09702	0.09702
	IAC Mucinous vs. IAC Papillary	0.35352	0.22480	0.23418	0.07300	0.20372	0.78462	0.78462
	IAC Mucinous vs. IAC Tubular	0.07514	**0.00634**	0.78488	**0.00846**	**0.01260**	0.92004	0.92004
	IAC Papillary vs. IAC Tubular	0.95976	0.97014	0.20794	0.84260	0.55078	0.79666	0.79666
TP53 mutation status	Mutant vs. Non-mutant	0.64328	0.28594	0.21586	**0.00600**	0.26196	0.35738	0.35738

**Table 3 ijms-25-08057-t003:** Expression levels of *miRNA-21-3p*, *miRNA-21-5p*, *miRNA-106a*, *miRNA-122-3p*, *miRNA-122-5p*, *miRNA-143-3p*, *miRNA-143-5p*, *miRNA-203a-3p*, *miRNA-203-5p miRNA-551b-3p*, *miRNA-551b-5p*, and *miRNA-574-3p* in each LAGC case before NAC (pre-surgery), and the correlation with responses to NAC. Responses to NAC were assessed by CT and HP. The responses were divided according to the dimension of the response (0—without response, 1—minor response, 2—major response, and 3—complete response). A *p* value of <0.05 (orange) was considered significant. miRNA, microRNA; LAGC, locally advanced gastric cancer; NAC, neoadjuvant chemotherapy; CT scan, computed tomography; and HP, histopathological.

Case Number	*miR-21**	*miR-21*	*miR-106a*	*miR-122*	*miR-122**	*miR-143*	*miR-143**	*miR-203a-5p*	*miR-203*	*miR-551b*	*miR-551b**	*miR-574-3p*	CT Response	HP Response
1.	*p* > 0.05	*p* > 0.05	*p* > 0.05	*p* > 0.05	*p* > 0.05	*p* > 0.05	*p* > 0.05	*p* > 0.05	*p* > 0.05	*p* > 0.05	**0.02**	*p* > 0.05	0	2
2.	*p* > 0.05	*p* > 0.05	*p* > 0.05	*p* > 0.05	*p* > 0.05	*p* > 0.05	*p* > 0.05	*p* > 0.05	*p* > 0.05	*p* > 0.05	*p* > 0.05	*p* > 0.05	1	3
3.	*p* > 0.05	*p* > 0.05	*p* > 0.05	*p* > 0.05	*p* > 0.05	*p* > 0.05	*p* > 0.05	*p* > 0.05	*p* > 0.05	*p* > 0.05	*p* > 0.05	*p* > 0.05	1	2
4.	*p* > 0.05	*p* > 0.05	*p* > 0.05	*p* > 0.05	*p* > 0.05	*p* > 0.05	*p* > 0.05	*p* > 0.05	**0.03**	*p* > 0.05	**0.01**	*p* > 0.05	1	1
5.	*p* > 0.05	*p* > 0.05	*p* > 0.05	*p* > 0.05	*p* > 0.05	*p* > 0.05	**0.04**	*p* > 0.05	**0.07**	*p* > 0.05	**0.00**	*p* > 0.05	1	1
6.	*p* > 0.05	*p* > 0.05	*p* > 0.05	*p* > 0.05	*p* > 0.05	*p* > 0.05	**0.02**	*p* > 0.05	*p* > 0.05	*p* > 0.05	**0.00**	*p* > 0.05	1	3
7.	*p* > 0.05	*p* > 0.05	*p* > 0.05	*p* > 0.05	*p* > 0.05	*p* > 0.05	*p* > 0.05	*p* > 0.05	*p* > 0.05	*p* > 0.05	*p* > 0.05	*p* > 0.05	2	3
8.	*p* > 0.05	*p* > 0.05	*p* > 0.05	*p* > 0.05	*p* > 0.05	**0.001**	*p* > 0.05	*p* > 0.05	*p* > 0.05	*p* > 0.05	*p* > 0.05	*p* > 0.05	2	1
9.	*p* > 0.05	*p* > 0.05	*p* > 0.05	*p* > 0.05	*p* > 0.05	*p* > 0.05	*p* > 0.05	*p* > 0.05	**0.01**	*p* > 0.05	*p* > 0.05	*p* > 0.05	3	3
10.	*p* > 0.05	*p* > 0.05	*p* > 0.05	*p* > 0.05	*p* > 0.05	*p* > 0.05	**6.36271 × 10^−5^**	*p* > 0.05	*p* > 0.05	*p* > 0.05	*p* > 0.05	*p* > 0.05	3	3

**Table 4 ijms-25-08057-t004:** Basic patient characteristics—cTNM, ycTNM, and ypTNM; T—tumor; N—lymph node, M—metastasis, c—clinical, yc—post-therapy clinical stages, and yp—post therapy pathological stages.

Variable		cTNM	ycTNM	ypTNM
Stage	T0	0	2 (20%)	5 (50%)
	T1	0	0	2 (20%)
	T2	1 (10%)	0	0
	T3	4 (40%)	4 (40%)	3 (30%)
	T4	5 (50%)	4 (40%)	0
Lymph nodes	N0	3 (30%)	6 (60%)	7 (70%)
	N1	3 (30%)	2 (20%)	2 (20%)

**Table 5 ijms-25-08057-t005:** Basic patient characteristics included in the study. n—number of cases, G—grade, FLOT—chemotherapy with docetaxel, oxaliplatin, and fluorouracil/leucovorin, and FLO—chemotherapy with 5-FU, leucovorin, and oxaliplatin.

Variable		All Patients (n = 10)
Age [mean]		61 years
Sex	Female	4 (40%)
	Male	6 (60%)
Localization	Corpus	5 (50%)
	Cardia	5 (50%)
Grade before chemotherapy	G1	1 (10%)
	G2	4 (40%)
	G3	5 (50%)
Grade after chemotherapy	G1	3 (30%)
	G2	1 (10%)
	G3	3 (30%)
	Gx	5 (50%)
Chemotherapy	FLOT	6 (60%)
	FLO	4 (40%)

## Data Availability

Data is contained within the article.

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
