# Peer review of "Diagnostic Potential of miR-143-5p, miR-143-3p, miR-551b-5p, and miR-574-3p in Chemoresistance of Locally Advanced Gastric Cancer: A Preliminary Study"

_ijms, 2024, doi:10.3390/ijms25158057_

Round 1

Reviewer 1 Report

Comments and Suggestions for Authors

This is a meaningful exploratory study that reveals the potential of some miRNA markers in the diagnosis and evaluation of chemotherapy response in gastric cancer. However, the research is still in its initial stages and requires further expansion of the sample size and more functional experiments to provide more sufficient evidence for clinical application.

I hope the authors can consider the following points:

1. The sample size is relatively small, with only 10 patients with locally advanced gastric cancer included, which may not be sufficient to represent a broader population.

2. There is a lack of control group data. Although the TCGA database was used for miRNA expression difference analysis, no control group without neoadjuvant chemotherapy was set up in the study samples, making it impossible to determine whether the changes in miRNA expression are specifically related to chemotherapy drug response.

3. The study primarily focuses on the expression levels of miRNAs but lacks an in-depth discussion of the specific roles of these miRNAs in the mechanism of chemotherapy resistance. Consider supplementing functional experiments, conducting more comprehensive bioinformatics analyses, and predicting and validating the target genes of miRNAs and their roles in chemotherapy resistance.

4. No long-term follow-up data of patients were provided, making it impossible to evaluate the relationship between miRNA expression levels and long-term treatment effects or survival rates.

5. The discussion on the potential of miR-551b-5p in assisting CT to evaluate the efficacy of chemotherapy is somewhat weak and lacks more evidence. It is recommended that the authors either supplement supporting data or weaken this argument.

6. The presentation of figures and tables may not be clear enough, and the interpretation of data may not be detailed enough.

7. There are still some deficiencies in the language expression of the article, and some sentences are not concise enough. It is suggested that the authors polish the language and logic of the entire text to improve the readability of the article.

Comments on the Quality of English Language

To ensure that your work meets the highest standards of scientific communication and readability, I strongly recommend engaging a professional third-party service for both scientific and language polishing.

Reviewer 2 Report

Comments and Suggestions for Authors

The article by Janiczek-Polewska M. et al. describes the diagnostic role of selected miRNAs in locally advanced gastric cancer. 

The abstract needs to be rewritten as it does not convey the most exciting part of the results. Also, while the authors mention the results of the analysis, it would be good to write about the details, as in this case: "The expression levels of miR-143-3p, miR-143-5p, miR-203a-3p and miR-551b-5p are increased in a few patients who responded to NAC". - The authors indicate that the fold change of selected miRNAs is increased, but do not indicate the level of change, which is crucial for the hypothesis of diagnostic biomarker potential. Furthermore, the authors should be very careful with the definition of "diagnostic biomarker" in their cases, where they only analysed the expression in ten patients. 

The introduction section lacks some information; as we know from the manuscript that "gastric cancer is the fifth most common cancer", we did not get any information on how many people are diagnosed annually. In addition, the introduction section did not contain the hypothesis of the studies, nor information on why selected miRNAs were chosen for further analysis. When the authors mentioned biomarkers (line 88), it is crucial to mention what kind of biomarkers they had in mind (diagnostic/prognostic/therapeutic).

The description of the discussion should be more transparent and synthetic to make it easier for reviewers to understand the content. In addition, some questions need improvement. Table 2 needs to include values expressed as 10^X, not 0.00008. In its current form, the table could be more readable. 

The titles of the subsections also need improvement. The figures (Figure 1) also need improvement and standardisation of scale. In its current form, it is difficult to read which gene's expression is higher and which is lower. In addition, the colours used in the figure should be more distinguishable.

The discussion should be rewritten, taking into account the analyses carried out and critically reviewing the conclusions drawn from them. In its current form, the discussion needs to be more precise in order to fully understand the results presented (and to indicate the biological significance of the miRNAs analysed). It should also indicate which information in the literature is consistent and which contradicts the results obtained. In addition, the Discussion section should include a section discussing the limitations of this study. The article contains minor linguistic errors. 

The conclusion should be rewritten. The authors cannot say that some miRNAs are potential diagnostic markers when they only performed the analysis on ten patients. Furthermore, the authors did not analyse circulating miRNA in the patients' serum. In this case, although biopsy is a gold standard for diagnosis, it also excludes the definition of diagnostic biomarkers. Therefore, the title of this article should be changed. 

In addition, some questions need to be asked:

- Why do the authors use U6 as a normalisation for their genes?

- After reading the whole article, I still do not fully understand why selected miRNAs were used for further analysis.

Comments on the Quality of English Language

Minor editing of English language required

Round 2

Reviewer 1 Report

Comments and Suggestions for Authors

While the authors have not fully addressed my initial concerns, this article still holds some value. However, for publication, two critical issues must be resolved:

1. The axis labels in Figures 3 and 4 are illegible. This must be corrected to ensure proper data interpretation.

2. The article is difficult to comprehend. The authors must engage a professional editing service to thoroughly revise the English throughout the manuscript.

These issues are non-negotiable and must be addressed before the paper can be considered for publication.

Comments on the Quality of English Language

The authors must engage a professional editing service to thoroughly revise the English throughout the manuscript.

Author Response

Thank you very much for your opinion of our article. We have improved the readability of Figures 3 and 4. We commissioned a language correction.  We used mdpi for language proofreading. We have a language proofreading certificate. We hope that the article will now be understandable and that it will be possible to accept it in this form.

Reviewer 2 Report

Comments and Suggestions for Authors

Accept in present form

Author Response

Thank you very much for your accepting of our article.